# Visible-light-driven reversible shuttle vicinal dihalogenation using lead halide perovskite quantum dot catalysts

Yonglong Li [1], Yangxuan Gao[1], Zhijie Deng[2,3], Yutao Cao[1], Teng Wang[1], Ying Wang[1], Cancan Zhang[1,4], Mingjian Yuan [1] & Wei Xie [1] ✉

Dihalogenation of alkenes to the high-added value vicinal dihalides is a prominent process in modern synthetic chemistry. However, their effective conversion still requires the use of expensive and hazardous agents, sacrificial half-reaction coupling or primary energy input. Here, we show a photocatalytically assisted shuttle (p-shuttle) strategy for redox-neutral and reversible vicinal dihalogenation using low-cost and stable 1,2-dihaloethane under visible light illumination. Energetic hot electrons from metal-halide perovskite QDs enable the challenging photocatalytic reactions. Ultrafast laser transient absorption spectroscopy have unveiled the energy matching of the hot electrons with the high reduction potential of 1,2-dihaloethane, via two consecutive photo-excitation process. Powered by the sustainable energy as the only energy input, our new catalytic system using metal-halide perovskite QDs for dibromination, dichlorination and even unexplored hetero-dihalogenation, shows good tolerance with a wide range of alkenes at room temperature. In contrast to homogeneous photocatalysts, chalcogenide QDs and other semiconductor catalysts, perovskite QDs deliver previously unattainable performance in photoredox shuttle vicinal dihalogenation with the turnover number over 120,000. This work provides new opportunities in visible-light-driven heterogeneous catalysis for unlocking novel chemical transformations.

From the viewpoint of synthetic chemistry, dihalogenation of alkenes is highly prevalent to construct two vicinal carbon–halogen bonds in one molecule[1,2]. In general, three strategies have been used (Fig. 1a): first, using hazardous $Cl_2$ and $Br_2$ as halogenation reagents[3,4]; second, using $X_2$ alternatives[5–7] such as $Et_4NCl_3$ and N-bromosuccinimide to release the corrosive $X_2$ in the reactions; Third, the combination of $X^-$ and a strong oxidant for in situ generating nucleophilic halogen sources[8,9]. The conventional strategies make dihalogenated processes costly and potentially hazardous[10]. Recently, some sustainable approaches have been developed in dihalogenation reactions using halogen ions, as illustrated by the electrocatalytic dihalogenation[11,12] and the photocatalytic dichlorination[13]. However, the reactions rely on oxidation-half-reactions of halogen ions accompanied by by-products generated in reduction-half-reactions to close the whole

[1]State Key Laboratory of Advanced Chemical Power Sources, Key Laboratory of Advanced Energy Materials Chemistry (Ministry of Education), Haihe Laboratory of Sustainable Chemical Transformations, Renewable Energy Conversion and Storage Center, College of Chemistry, Nankai University, Tianjin 300071, P. R. China. [2]State Key Laboratory of Elemento-Organic Chemistry, College of Chemistry, Nankai University, Tianjin 300071, P. R. China. [3]Mount Sinai Center for Therapeutics Discovery, Departments of Pharmacological Sciences, Oncological Sciences and Neuroscience, Tisch Cancer Institute, Icahn School of Medicine at Mount Sinai, New York, NY 10029, USA. [4]State Key Laboratory of Marine Resource Utilization in South China Sea, Hainan Provincial Key Laboratory of Research on Utilization of Si-Zr-Ti Resources, College of Materials Science and Engineering, Hainan University, Haikou 570228, P. R. China. ✉e-mail: wei.xie@nankai.edu.cn

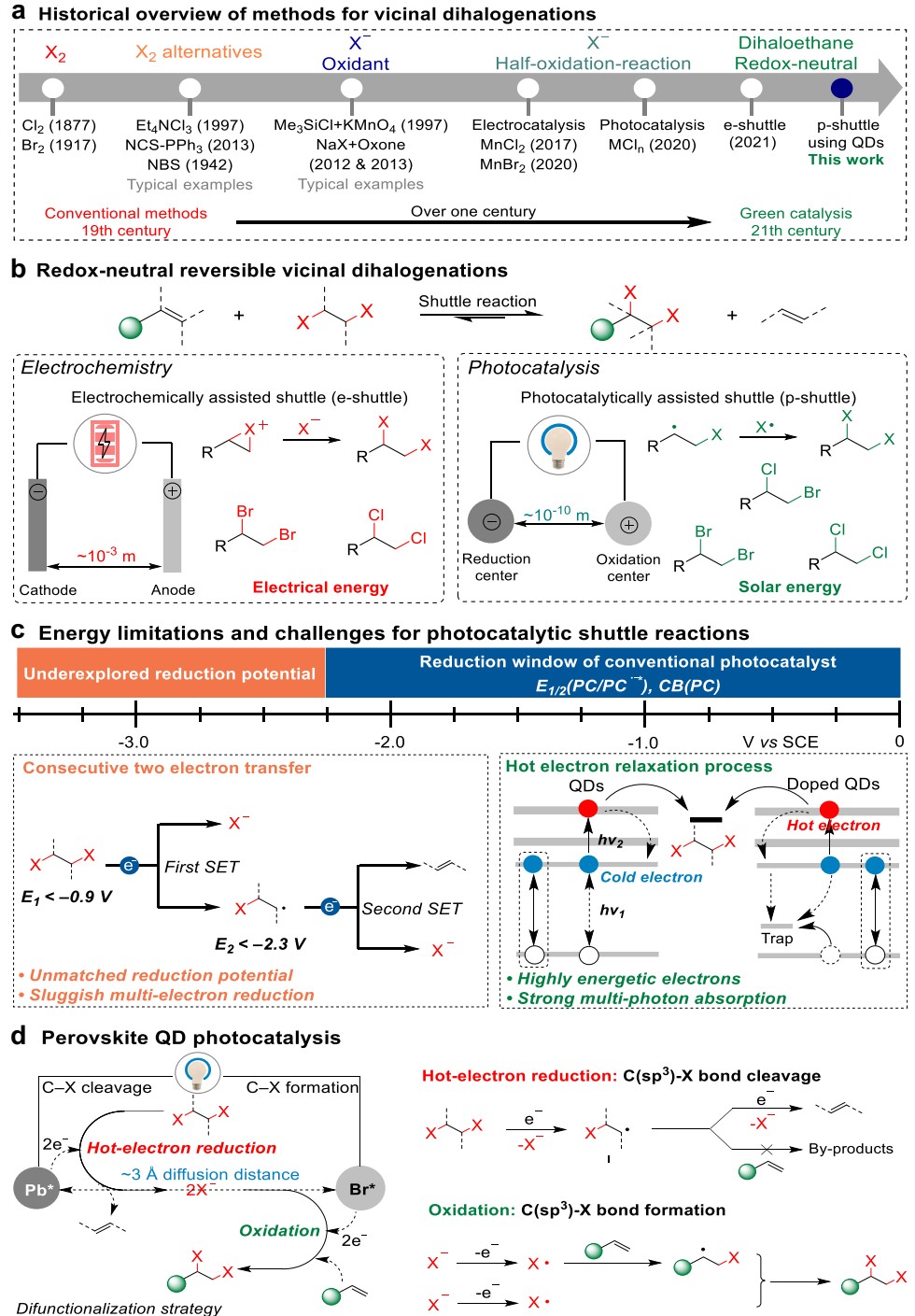

**Fig. 1 | Vicinal dihalogenation of alkenes. a** Historical methods for vicinal dihalogenations. **b** Electrocatalysis- vs. photocatalysis-enabled redox-neutral shuttle reactions. **c** Energy limitations for photoelectric conversion catalysis and challenges in the development of photocatalytic transfer difunctionalization. **d** P-shuttle reactions on perovskite QD surface.

electrocatalytic or photocatalytic process. A functional group transfer methodology could be a very desirable strategy[14,15]. In 2021, a particularly elegant proof-of-principle example, consecutive paired electrolysis enables a reversible, electrochemically assisted shuttle dihalogenation transformation (e-shuttle, Fig. 1b) using 1,2-dihaloethane, was reported by Morandi group[10]. Despite this great success in the redox-neutral dihalogenation, this reaction, as well as many other electrochemical reactions, has many inherent limitations including highly intensive primary energy input[16], overoxidation of molecular substrates[17,18] and challenging removal of electrolytes after reaction[19,20].

Compared with the electrosynthesis, visible-light-driven photoredox catalysis has attracted substantial attention for accessing efficient and cost-effective chemical transformations due to the green energy (solar energy) input[21–26]. However, the formidable challenge in developing transfer dihalogenations using visible-light photocatalysis originates from the redox potentials bounded by the energy of photons[27,28], because 1,2-dihaloethane possesses a higher energy level

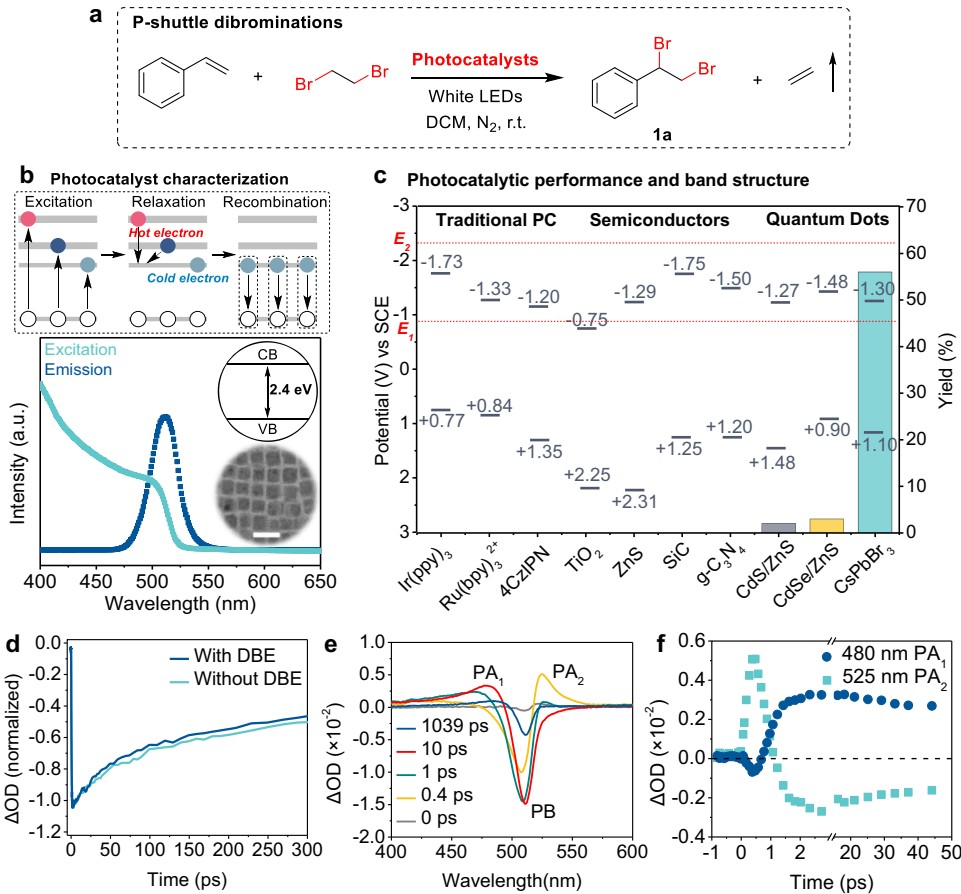

**Fig. 2 | Hot electron-induced shuttle dibromination reactions. a** Visible-light-driven dibromination reactions. **b** CsPbBr₃ QD photocatalyst characterization. Schematic carrier relaxation, excitation (green lines), emission spectra (blue lines), bandwidth, and TEM image (scale bar: 20 nm) of CsPbBr₃ QDs. **c** Photocatalytic performance and band structure of various photocatalysts. **d** Transient dynamics of CsPbBr₃ QDs with and without DBE. **e** Time-resolved transient absorption spectra of CsPbBr₃ QD catalysts. **f** Transient dynamics of the CsPbBr₃ QD PA₁ at 480 nm and PA₂ at 525 nm.

for its lowest unoccupied molecular orbital (LUMO) than the conduction band minimum (CBM) of conventional photocatalysts (Fig. 1c).

Under visible light stimulation, the utilization of hot electrons generated from semiconductor quantum dots (QDs)[28–33] or plasmonic metal nanoparticles[34–38] holds great promise in breaking through the restriction of conventional photocatalysis. This is due to the unique properties of these hot electrons, which possess excess kinetic energy. Their high kinetic energy enables hot electrons to exhibit favorable reduction potential to overcome energy barriers for efficient charge transfer, ultimately leading to energetically demanding photocatalytic reactions[28,31,32]. In the case of QDs, hot electrons with energy levels above the band gap are generated through the interaction of two photo-excited excitons. These energetic electrons exhibit potentials several volts higher compared to typical hot electrons decayed from metallic surface plasmons through single-photon processes[33]. Hot electrons in QDs produced via exciton-to-hot-electron upconversion possess large excess energy that leads to the average potential level < −3.5 V with respect to a saturated calomel electrode (vs. SCE)[33]. This value is more negative than the reduction potential value of metallic Li⁺ ($E_{red}$ = −3.29 V vs. SCE). Recently, some studies reported lead halide perovskite QDs hold more tremendous potential than conventional QDs in generating long-lived hot carriers[39,40]. While only the cold carriers on the band gap of perovskite QDs are being explored for photoredox organic synthesis including C−X, C−C, C−O, C−N, S−S, and N−N bond formation, and also C−H functionalization reactions[41–47], the utilization of the hot carriers from perovskite QDs for highly efficient photocatalytic organic reactions, specifically the reactions cannot be driven by cold carriers, has remained unexplored.

Here, we show the energetic hot electrons from perovskite QDs are efficiently utilized to trigger redox-neutral shuttle vicinal dihalogenation under visible light illumination (Figs. 1b and c). The dihalogenation involves a reduction–oxidation cascade—the reversible transformation between two highly stable carbon-halogen bonds between safe and inexpensive 1,2-dihaloethane and highly valuable vicinal dihalide through a two-electron-transfer process. During the photoredox reaction, two half-reactions occur at the same QD catalyst[48]. The dihaloethane undergoes reduction on the electron-rich Pb center (Pb*, Fig. 1d) to release a first X⁻ anion and a carbon radical I. Through consecutive hot electron transfer, radical I interacts with the surface Pb* again and generates a second X⁻ and an alkene. As a central step in the reaction, CsPbBr₃ QDs generate hot electrons to break two carbon–halogen bonds and effectively suppress the competing undesired coupling between the alkene substrate and carbon radical I. Subsequently, active X⁻ species generated on the reducing site (Pb*) spill over ~10⁻¹⁰ m distance to a oxidizing site (Br*)[41]. Two X⁻ anions are almost simultaneously oxidized at the Br* site (h⁺) to produce the corresponding radicals, followed by combination with the alkene to reestablish carbon-halogen bonds. This is a rare example of exploring excited hot electrons in perovskite QDs for driving chemical reactions.

## Results

### Photocatalytic shuttle dibrominations using CsPbBr₃ QDs

The experimental section provided the synthesis details of colloidal CsPbBr₃ QDs (Supplementary Table 1). A transmission electron microscopic image reveals that the cubic-shaped CsPbBr₃ QDs have an average edge length of 9.8 ± 0.6 nm. Fig. 2b displays the extinction and

emission spectra of the CsPbBr$_3$ QDs, which demonstrates the lowest energy excitonic transition bands of the QDs is located at 518 nm (corresponding band energy of 2.4 eV). When the prepared QDs are excited by 400–500 nm visible light, in a typical process, energetic hot electrons are generated and then quickly equilibrated with the lattice temperature to release energy via scattering phonons at sub-ps time-scale. Subsequently, the recombination of cold excitons on the bandgap leads to the photoluminescence (PL) emission (Fig. 2b). It was originally hypothesized that extracting hot carriers from CsPbBr$_3$ QDs to substrate molecules would significantly improve the photocatalytic efficiency or enable energetically demanding photoredox conversions[40]. To demonstrate the substantial ability of hot electrons to trigger photoredox reactions, an easily accessible quartz setup charged with 0.1 mmol of alkene, 0.5 mmol of 1,2-dibromoethane (DBE), and 0.00045 mol% of CsPbBr$_3$ QDs in dichloromethane (DCM) solvent is irradiated with a 10 W white light-emitting diode (LED, Supplementary Fig. 1) under nitrogen atmosphere. Upon the 400–750 nm continuous visible-light stimulation, energetic charge carriers are formed and distributed at the corresponding redox sites, which results in the formation of desired product (1,2-dibromoethyl) benzene (**1a**) with 65% yield and the release of gaseous product ethylene under 25 °C (Fig. 2a). The gas-phase reaction mixture is characterized using gas chromatography–mass spectrometry (GC–MS), while the liquid-phase reaction mixture is analyzed using $^1$H-nuclear magnetic resonance ($^1$H-NMR, see Supplementary Figs. 2 and 3). The results reveal that the formed ethylene is released from the liquid solution to gas phase, providing a driving force for the shuttle process. The colloidal QDs are easily separated from the reaction mixture via centrifuging to be recycled (Supplementary Fig. 4). Control reactions confirm that no reaction occurs without QD catalysts or without visible-light irradiation (Supplementary Table 2). The reversibility of the p-shuttle dibromination is demonstrated in Supplementary Fig. 5.

The related cost in our optimized p-shuttle dibromination reactions is extremely low (down to 2.25 $/mmol), one order of magnitude lower than in the reported e-shuttle strategy (51.99 $/mmol, Supplementary Fig. 6). We also demonstrate the CsPbBr$_3$ QD-catalyzed disbrominations with an unprecedented combination of performance parameters: turnover number (TON) to be at least 342,000 and quantum yield (QY) to be about 3.6% (details in the Supplementary Information). Chemical transformations driven by visible light were often associated with the use of homogeneous redox photocatalysts[23] (such as transition-metal Ru(II) and Ir(III) based complexes and organic dyes), heterogeneous semiconductors[24,49] (such as TiO$_2$, ZnS, SiC, and g-C$_3$N$_4$), or chalcogenide QDs[50] (such as CdS/ZnS and CdSe/ZnS). However, under identical conditions, no or trace yield of **1a** can be obtained using 9 other different photocatalysts (Fig. 2c and Supplementary Table 3). Based on the results obtained from the optimization experiments (Supplementary Fig. 7 and Tables 4–8), CsPbBr$_3$ QD catalysts exhibit good performance in the vicinal dibromination. Higher activity of CsPbBr$_3$ QDs than other photocatalysts may originate from the intrinsic photophysical properties on hot carriers production, relaxation, and transfer[51,52]. According to recent reports, lead halide perovskite QDs can produce long-lived hot carriers, because the highly dynamic crystal structure and strong quantum confinement effect effectively suppress the strong carrier-phonon coupling to slow hot carriers cooling rates[53–55].

Because DBE molecule ($E_1 = -0.87$ V and $E_2 = -2.32$ V vs. SCE)[10] possesses a higher reduction potential than the CBM of QDs ($V_{CB} = -1.3$ V vs. SCE), only hot electrons with high energy levels above the band-edge, characterized by $E < -2.3$ V vs. SCE, can be transferred to DBE driven by a favorable energy matching. To better understand the process of extracting hot electrons from CsPbBr$_3$ to DBE, we performed ultrafast laser time-resolved transient absorption spectroscopy (TAS). Figure 2d shows a faster decay of the photoinduced bleach (PB) of TA kinetics at the 510 nm excitation band of the CsPbBr$_3$

QDs/DBE mixture. The results indicate that the photo-generated electrons in QD catalysts are initially transferred into the antibonding orbital of DBE, which leads to an accelerating PB recovery. Continuous change of emission spectra (Supplementary Fig. 8) under light irradiation, also reflects that the DBE extracts photo-generated electrons from QDs during the photoredox process.

The TA spectra of CsPbBr$_3$ QDs in the presence of DBE (Fig. 2e) show two well-resolved bands of photoinduced absorptions: the first one (PA$_1$) is centered at 480 nm, the second one (PA$_2$) is centered at 525 nm[56]. The two bands are both attributed to the transitions from CBM states to the states above band-edge upon second photon stimulation, leading to the generation of hot electron with a high reduction potential. At very short delay (500 fs), the difference in optical density (ΔOD) of the PA$_2$ increases to the maximum value and then the band also undergoes fast decay (500 fs), indicating that the lifetime of generated hot carrier driven by visible light of 525 nm is very short (<1 ps). However, the PA$_1$ relaxation process exhibits a two-step distinct dynamic decay, with a fast increasing trend of ~2 ps, and followed by a subsequent slower and gradual decay lasting several tens of ps (Fig. 2f). The long lifetime of PA$_1$ allows hot electrons transfer to the surroundings of the QDs. According to the band structure of CsPbBr$_3$ QDs, the potential of free electrons on the band edge is located at around −1.3 V vs. SCE, which is not sufficient to induce consecutive electron transfer to DBE ($E_2 = -2.32$ V vs. SCE)[10] to activate the two-electron reduction process. Hot electrons are excited by the second photon (2.6 eV), corroborating the PA$_1$ feature within the spectral range approximately centered at 480 nm and are thermodynamically competent of injection into DBE to enable the second C–Br bond cleavage and generate Br$^-$ anion. The subsequent oxidation of two Br$^-$ anions on the QD surface (Supplementary Fig. 9) leads to the formation of corresponding radicals, which then react with the alkene to generate the dibrominated product, thus closing the p-shuttle process.

## Scope of photocatalytic shuttle dibrominations

Using the energetic hot electrons within perovskite QDs, we convert a broad range of alkenes to vicinal dibromides (**1a** to **29a** in Fig. 3). Under visible light illumination at room temperature, various aryl-substituted alkenes undergo smooth conversion to the corresponding vicinal dibromides in high yields (**1a** to **8a**). The scope of this perovskite QD-catalyzed dibromination reaction has been extended to classes of other substrates beyond styrenes (**9a**). Notably, 1,1-disubstituted (**10a**) and 1,2-disubstituted (**11a** to **14a**) alkenes are used as substrates as well to produce the corresponding 1,2-dibromides in modest yields via radical dibromination. The tolerance of a large variety of functional groups under this photocatalytic conditions are investigated. The mild conditions required for the desired difunctionalization using CsPbBr$_3$ or Cu-CsPbBr$_3$ QDs (vide infra) shows an excellent chemoselectivity profile. As such, substrates containing CO$_2$Me, CN, CF$_3$, NO$_2$, TMS, Bpin, CHO, and heterocycle groups are all proved well compatible with the visible-light-driven reaction system (**15a** to **19a**, **22a to 29a** in Fig. 3). To our delight, the more challenging study to unlock p-shuttle dibromination of alkynes is realized (**20a** and **21a**).

## Photocatalytic shuttle dichlorinations

To explore a transfer dichlorination reaction, we choose 1,1,2,2-tetra-chloroethane (TCE, $E_1 = -2.23$ V)[10,57] as the chlorination source, because of its lower reduction potential compared with 1,2-dichloroethane ($E_1 = -2.79$ V)[10,57]. The photocatalytic dichlorination experiments are carried out using styrene as the archetypal substrate, CsPbBr$_3$ QDs as the photocatalyst, and acetonitrile (MeCN) as the solvent instead of DCM in dibromination protocol in order to exclude another chlorination source. However, only 5% of 1,2-dichloride **1b** is detected by GC, despite of a > 95% consumption of styrene (see Route II in Fig. 4a). A 1,1,2-trichloroethane-centered radical generated by the

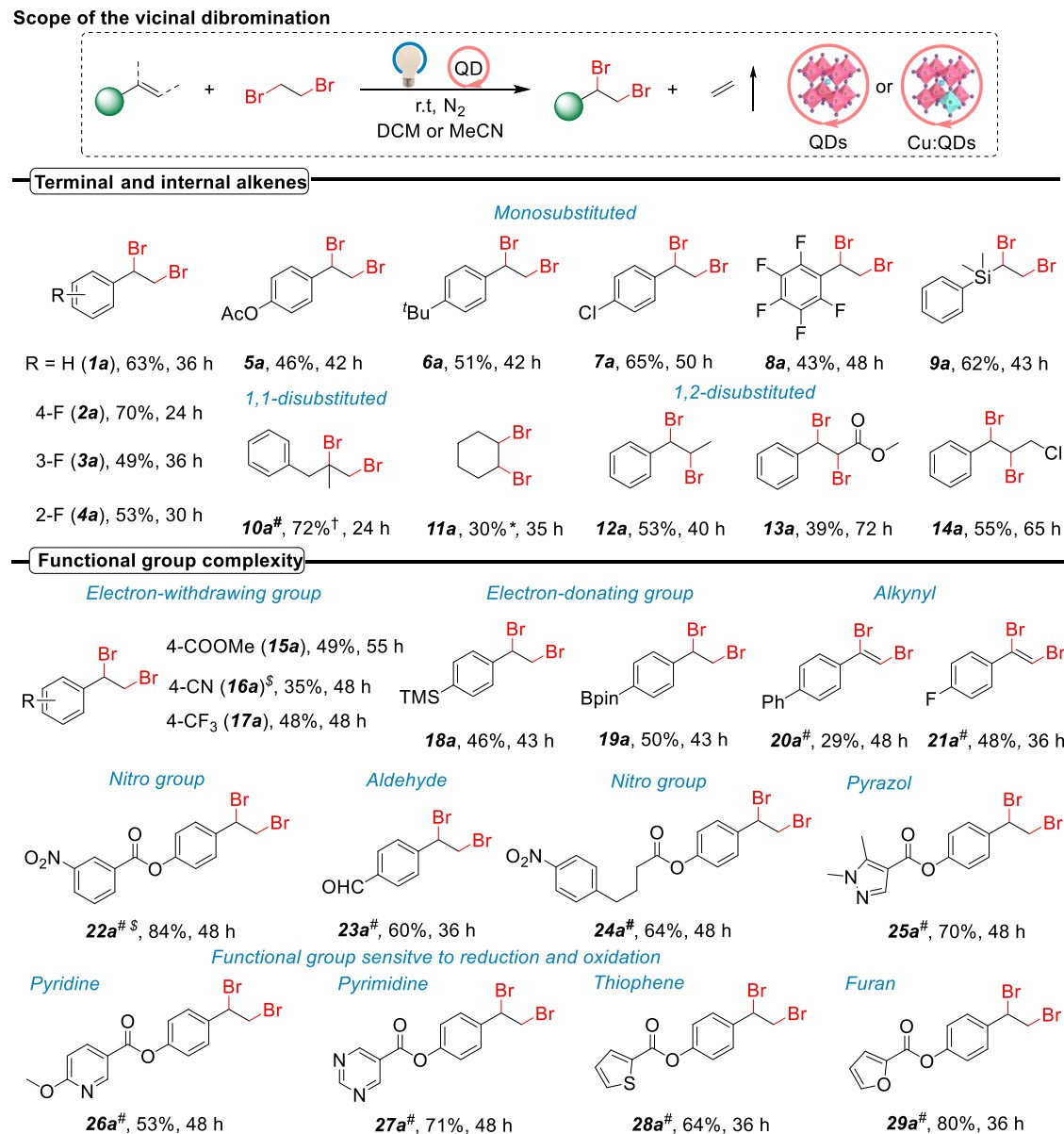

**Fig. 3 | Scope of alkene of the photocatalytic shuttle dibromination reactions.** All yields are isolated yields of the products unless otherwise noted. *GC yield with n-dodecane as the internal standard. †¹H-NMR yield. Reaction conditions: 0.1 mmol alkene, 0.5/0.75 mmol DBE, 4 mg CsPbBr₃ QD catalyst, 1 mL of DCM and 10 W white LED irradiation with continuous stirring at 25 °C. #0.5 mmol DBE, 4 mg Cu-CsPbBr₃ QD catalyst and 1 mL of MeCN. $0.75/1 mmol 1,1,2-tribromoethane.

first single-electron reduction of TCE can undergo an undesired transformation with styrene to yield two by-products **1b'** via an atom transfer radical addition (ATRA) reaction (Supplementary Fig. 10). In QD-catalyzed dichlorinations, the 1,1,2-trichloroethane-centered radicals fail to extract the second hot electron from CsPbBr₃ QDs to further release a second Cl⁻ anion, which leads that the second hot electron transfer process cannot outpace competing ATRA side reactions.

To accelerate the second hot electron transfer for the formation of two C−Cl bonds, we seek to utilize a doping metal ion to prolong the hot electron lifetimes of CsPbBr₃ QDs and enable the consecutive two-electron transfer. Doping of a metal ion in semiconductor QDs can introduce an intragap d state into the host QDs, which leads to ultrafast capturing of the photogenerated hole[58]. If the process of capturing holes is much faster compared to the relaxation of hot electrons, it will result in slow hot electron cooling in the system due to the decoupling of the photoinduced electron−hole (Fig. 4b)[58]. Hence, a series of metals such as copper, manganese, nickel, and silver are incorporated into CsPbBr₃ QDs by a cation-exchange method in order to prolong hot

electron cooling times. After several experiments, Cu-doped CsPbBr₃ QDs (Cu to Pb ratio of 3.0–7.0%) show fairly high catalytic activity in the visible-light-driven shuttle dichlorination (Supplementary Fig. 11 and Tables 9 and 10). The corresponding characterization data for Cu-CsPbBr₃ QDs (using CuBr₂ as metal source) are provided in Supplementary Fig. 12. A sustainable dichlorination protocol is developed using Cu-CsPbBr₃ QDs (0.00049 mol%) as the photocatalyst, TCE as the halogen source and MeCN as solvent, under nitrogen atmosphere at 25 °C and illuminated by a 10 W white LED (Supplementary Tables 11 and 12). 1,2-dichloride **1b** is obtained in 60% yield (with TON > 120,000), which is improved by ~12 times compared with that using the undoped CsPbBr₃ QDs (~5% yield) as photocatalysts. Crude ¹H-NMR result of the dichlorination reaction reveals the presence of 1,2-dichloroethylene as a by-product (Supplementary Fig. 13). Using this mild photocatalytic dichlorination, various alkenes (**1b** to **12b** in Fig. 4c) are smoothly converted to the targeted dichlorinated product in moderate yields. Moreover, Cu-CsPbBr₃ QDs as photocatalysts can also effectively catalyze the vicinal dibromination (**25a** to **29a** in Fig. 3).

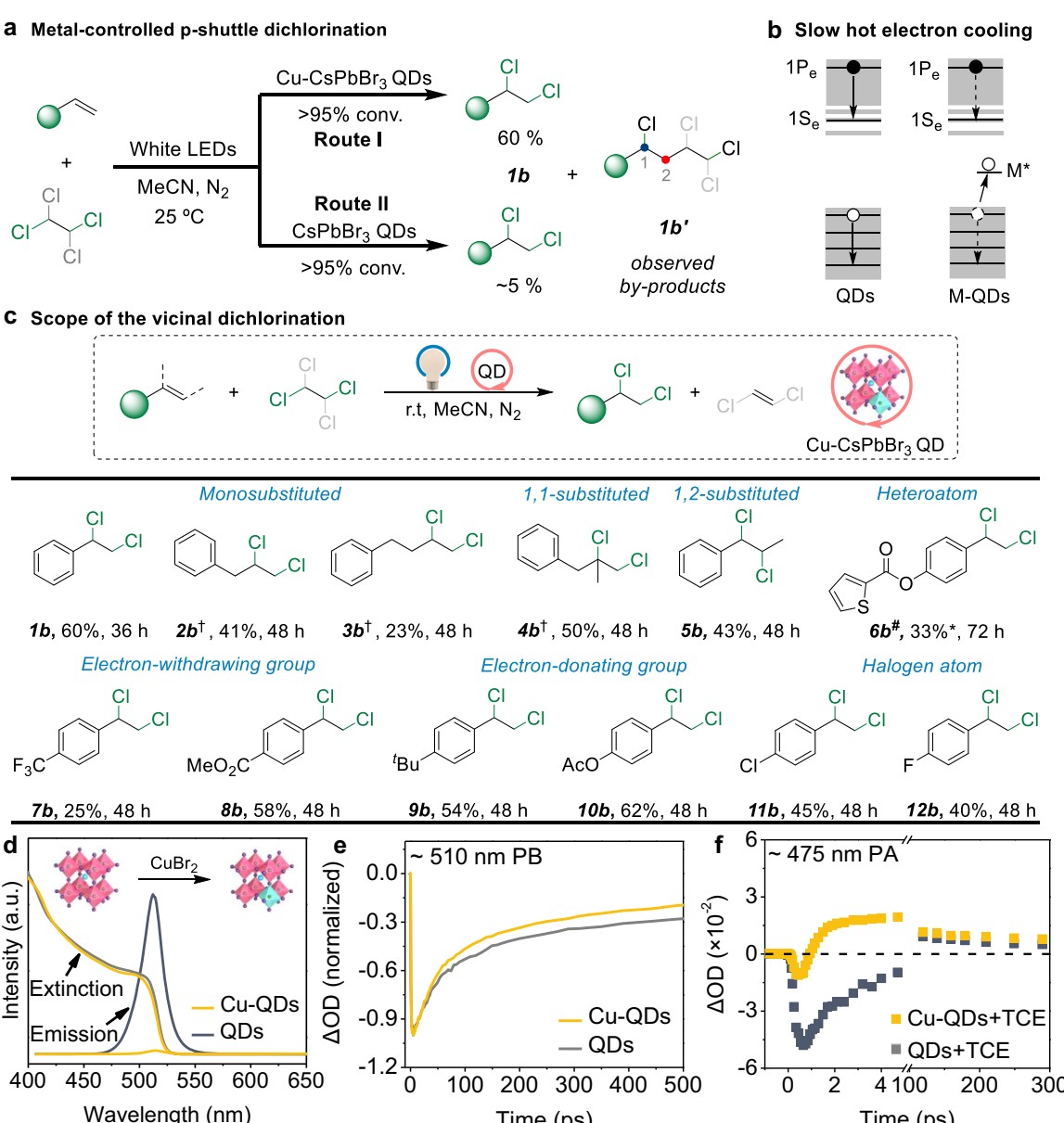

**Fig. 4 | P-shuttle dichlorination reactions via metal-controlled hot electron generation. a** Schematic representation of metal-controlled p-shuttle dichlorination. **b** Slow hot electron cooling in metal-doped QDs. After using metal dopants, the ultrafast capture of holes by the intragap states leads to the suppression of the hot electron cooling process. **c** Scope of alkene of the photocatalytic vicinal dichlorination. All yields are GC yields of the products unless otherwise noted.

*[1]H-NMR yield. Reaction conditions: 0.1 mmol alkene compound, 1.35 mmol TCE, 4 mg Cu-CsPbBr$_3$ QD catalyst, 1 mL of MeCN, and irradiation using a 10 W white LED at 25 °C. [†]1.93 mmol TCE as the donor.[#]4 mL of MeCN as solvent. **d** Emission and absorption spectrum for CsPbBr$_3$ QDs, treated with CuBr$_2$. **e, f** Transient dynamics of the CsPbBr$_3$ and Cu-CsPbBr$_3$ PB at 510 nm (**e**) and PA at 475 nm (**f**).

After Cu doping, the extinction spectrum of the QDs shows no obvious change, but an immediate quenching is observed in the PL emission curve (Fig. 4d). These results suggest that Cu ions act as additional recombination centers to accelerate nonradiative relaxation of the charge carriers. In Fig. 4e, the Cu-QDs exhibit a generally faster bleach recovery compared to the undoped QDs, which can be attributed to the localization of holes at the Cu sites[46]. Because the holes are quickly captured by Cu sites, the electrons on the 1Se level can be excited by a second photon to a higher energy level. In the presence of TCE, the PA relaxation process of Cu-doped QDs exhibits an advantageous decay of several tens of ps at the spectral window around 475 nm (Fig. 4f). Therefore, more hot electrons generated from Cu-doped QDs can be transferred into the LUMO of TCE to release two Cl$^-$ anions, which are subsequently oxidized into the radicals and trapped

by the alkene to reestablish the two C−Cl bonds (Supplementary Fig. 14).

## Photocatalytic shuttle hetero-dihalogenation
Motivated by the successful dibromination and dichlorination transformations, we proceed to expand the hot electron-induced shuttle reaction to hetero-dihalogenation. The precedent works of preparing hetero-dihalogenated molecules mainly rely on the use of BrCl or the combination of an electrophilic halogen reagent X$^+$ and a nucleophilic anion X$^-$ [59]. We wonder whether the p-shuttle strategy via halogen atom radical route with cheap and easily available DBE and TCE as halogen sources is feasible. The challenge arises predominantly from low functionalization sequentiality because of the formation of four possible difunctionalization products (Fig. 5a).

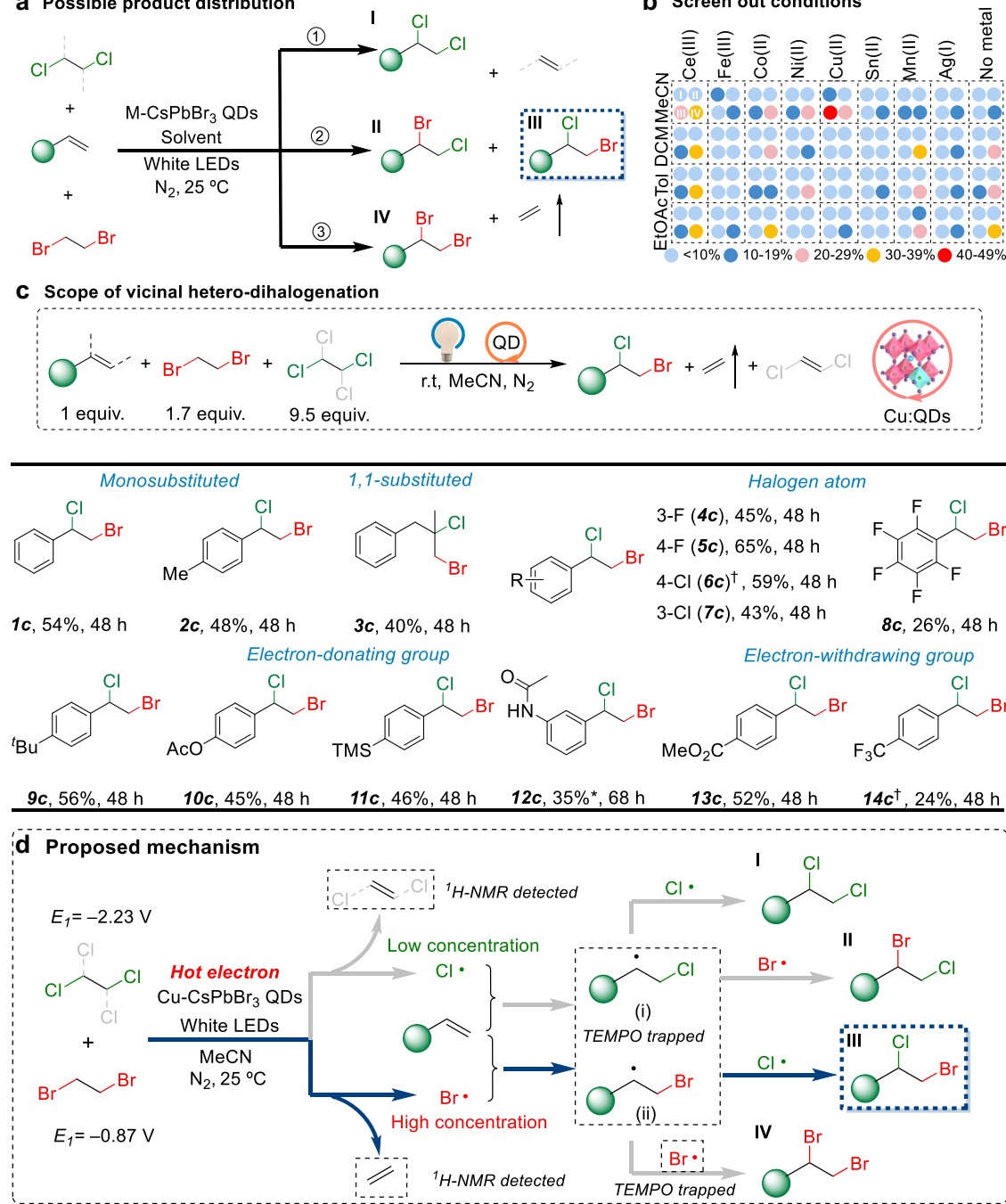

**Fig. 5 | Hetero-dihalogenation reactions via selective p-shuttle catalysis.**
**a** Possible product distribution. Four different products were generated via hetero-dihalogenation shuttle catalysis reactions. **b** Screen out different metals and solvents for optimizing the hetero-dihalogenation reaction conditions. **c** Scope of alkene of the photocatalytic hetero-dihalogenation. All yields are GC yields of the products unless otherwise noted. *[1]H-NMR yield. Reaction conditions: 0.1 mmol alkene compound, 0.17 mmol DBE, 0.95 mmol TCE, 5 mg Cu-CsPbBr₃ QD catalyst, 1 mL of MeCN and irradiation using a 10 W white LED at 25 °C. †0.2 mmol alkene compound as the substrate. **d** Proposed mechanism of the generation of hetero-dihalogenation via metal-controlled hot electron reduction.

To realize the efficient hetero-dihalogenation shuttle, a series of metal-doped CsPbBr₃ QDs are prepared to execute photocatalytic performance at 25 °C in a reaction vial with styrene as the halogen atom receptor, DBE and TCE as the halogen sources (Fig. 5b and Supplementary Table 13). It is noteworthy that the yield of dibromination product IV has dominated the products (Fig. 5b), even though using different catalyst-solvent combinations. The higher yield of dibrominated products should result from the heavy concentration of bromide radicals via the redox cascade of DBE. To be out of the dilemma, we utilize Cu-doped CsPbBr₃ QDs to greatly enhance the concentration of chlorine radicals via a continuous hot electron transfer, which leads to the selective formation of hetero-dihalogenated product with ~50% yield (Supplementary Fig. 15). The p-shuttle hetero-dihalogenation unlocks a series of substrates with different functional groups under visible light illumination (**1c** and **14c** in Fig. 5c), which is an unexplored strategy in the e-shuttle protocol[10].

We further aim to give mechanistic insight into the origin of shuttle hetero-dihalogenation catalyzed by Cu-doped QDs. The addition of 2,2,6,6-tetramethylpiperidine-1-oxyl (TEMPO), a radical trapping agent, is found to inhibit the formation of 4 difunctionalized products and

high-resolution mass spectrometry indicated that the key radical species (such as bromine and the C-centered radicals) are trapped by the agent (Supplementary Fig. 16). Under light irradiation, halide anions (Br⁻ and Cl⁻) are detected by emission spectroscopic characterization, which excludes the possibility that halogen agents directly produce halogen radicals (•Br and •Cl) via energy transfer (Supplementary Fig. 17). Hence, a plausible mechanism about selective hetero-dihalogenation is presented in Fig. 5d. During the photoredox process, DBE ($-0.87$ V vs. SCE) and TCE ($-2.23$ V vs. SCE) successively extract two photo-generated hot electrons within QDs and produce halide anions and the corresponding alkene (ethylene and 1,2-dichloroethylene), as confirmed by $^1$H-NMR analysis (Supplementary Figs. 18 and 19). The oxidation of Br⁻ ($E_{ox} = 0.75$ V vs. SCE) and Cl⁻ anions ($E_{ox} = 1.05$ V vs. SCE) can occur at the surface oxidation site, resulting in the formation of the corresponding radicals (•Br and •Cl)[41]. Subsequently, the radicals can rapidly add to the alkene to produce the C-centered radicals i and ii. Because of their lower oxidation potentials, the Br⁻ anions are more easily oxidized at the QD surface to generate •Br radicals with higher concentrations and result in the formation of dominant C-centered radicals ii. With the depletion of •Br radicals, radical ii is attracted by the •Cl radicals and converted to product III (Fig. 5d).

## Discussion

This work accesses the first-ever visible-light-driven vicinal dihalogenation of alkenes via a photocatalytic shuttle strategy using perovskite QDs, which deliver particularly high TON for the reactions. Compared with the conventional photocatalysts, the CsPbBr₃ QD catalyst generates energetic hot electrons to overcome the limitation of mismatched thermodynamics and sluggish kinetics of two-electron dibromination. Surface modification of the perovskite catalyst with Cu ions to slow down the cooling process of hot electrons and match the frontier orbitals of other 1,2-dihaloethane further expands the strategy to shuttle dichlorination and hetero-dihalogenation. This p-shuttle strategy using hot electrons for vicinal dihalogenation offers atomic economy and low-energy-input synthesis without the need for high-energy electricity input. Further investigations of other metal ions might upgrade the importance and use of the perovskite catalysts in chemical transformations.

## Methods

### Synthesis of CsPbBr₃ QD photocatalysts

CsPbBr₃ QDs were synthesized using a modified method based on our previous work[41]. First, preparation of Cs-oleate precursor solution and hot-injection of the precursor into PbBr₂ solution at 170 °C; Second, rapid cooling for the nucleation and QD growth within several seconds. To prepare the precursor solution, a 50 mL three-neck round-bottomed flask was charged with 0.2035 g of Cs₂CO₃, 1 mL of oleic acid (OA), and 10 mL of 1-octadecene (ODE). The solution underwent degassing at 100 °C under vacuum for 10 min and then flushed with nitrogen flow at 120 °C for 30 min. In another 50 mL three-neck flask, 0.2070 g of PbBr₂ and 15 mL of ODE were loaded and heated to 120 °C under nitrogen flow. After approximately 45 min, 1.5 mL of oleylamine (OAm) and 1.5 mL of OA were added to the PbBr₂ solution at 120 °C. The temperature of the mixture was elevated to 170 °C, followed by the rapid injection of 1.4 mL of Cs-oleate precursor using a glass syringe under a nitrogen flow. After a 5-s interval, the reaction was promptly halted by immersing it in an ice-water bath. The QDs were then separated from the solution through centrifugation at 8601$g$ and washed using a mixed solvent of ethyl acetate and hexane in a ratio of 2:1. The resulting perovskite QDs were subsequently re-dispersed in hexane for future applications.

### Synthesis of Cu-CsPbBr₃ QD photocatalysts

For the preparation of Cu-doped CsPbBr₃ QDs, a procedure was adapted from literature with slight modifications[46]. 0.005 mmol of CuBr₂ precursors were introduced to a suspension of 20 mg of CsPbBr₃ QDs in 1 mL of hexane, and the mixture solution was stirred in the absence of light for approximately 90 min. After then, the resulting solution was centrifuged, and the colloidal Cu-CsPbBr₃ QDs were obtained after removing CuBr₂ precipitation. The prepared Cu-CsPbBr₃ QDs were then added to a quartz tube and dried at room temperature for subsequent photocatalysis.

### Photocatalytic dibromination of alkenes using 1,2-dibromoethane (general procedure 1)

A 10 mL quartz tube was equipped with a stir bar and then filled with the corresponding alkenes (0.1 mmol, 1.0 equiv.), 1,2-dibromoethane (0.50/0.75 mmol, 5.0/7.5 equiv.), and 4 mg of CsPbBr₃ QDs. Then, 1 mL of dichloromethane was added. The solution mixture was subjected to sonication and then introduced into a nitrogen atmosphere using a syringe needle connected to a dual vacuum pump. The photocatalytic reaction was carried out under the irradiation of a 10 W white LED, providing a power intensity of approximately 165 mW/cm² within the wavelength range of 400–700 nm. To maintain a reaction temperature of 25 °C, an external thermostat was employed. For substrates with low reactivity, the photocatalytic reaction was conducted under the same conditions but using 1,1,2-tribromoethane as the donor. After the respective reaction time, the perovskite QDs were separated from the reaction mixture by centrifugation. The residue was further purified through column chromatography to yield the desired product.

### Photocatalytic dibromination of alkenes using 1,2-dibromoethane (general procedure 2)

A 10 mL quartz tube was equipped with a stir bar and then filled with the corresponding alkenes (0.1 mmol, 1.0 equiv.), 1,2-dibromoethane (0.50 mmol, 5.0 equiv.), and 4 mg of Cu-CsPbBr₃ QDs. Then, 1 mL of acetonitrile was added. The solution mixture was subjected to sonication and then introduced into a nitrogen atmosphere using a syringe needle connected to a dual vacuum pump. The photocatalytic reaction was irradiated by a 10 W white LED and cooled to maintain the reaction temperature (at 25 °C). After the respective reaction time, the subsequent steps followed the same procedure as in **general procedure 1**.

### Photocatalytic dichlorination of alkenes using 1,1,2,2-tetrachloroethane (general procedure 3)

A 10 mL quartz tube was equipped with a stir bar and then filled with the corresponding alkenes (0.1 mmol, 1.0 equiv.), 1,1,2,2-tetrachloroethane (1.35/1.93 mmol, 13.5/19.3 equiv.), and 4 mg of Cu-CsPbBr₃ QDs. Then, 1 mL of acetonitrile was added. The solution was subjected to sonication and then introduced into a nitrogen atmosphere using a syringe needle connected to a dual vacuum pump. The photocatalytic reaction was irradiated by a 10 W white LED and cooled to maintain at 25 °C. Afterward, for GC quantitative analysis, a specific quantity of n-dodecanes was added as internal standard compounds, along with 1 mL of dichloromethane as the solvent. Alternatively, for $^1$H-NMR quantitative analysis, a specific quantity of mesitylenes was utilized as internal standard compounds. The mixture was then subjected to centrifugation at 7607$g$ to remove the QD precipitates. Subsequently, the resulting solution was filtered and subjected to analysis using $^1$H-NMR, GC, and GC–MS techniques. The quantitative analysis of each product was conducted by referring to GC calibration curves established with standard chemicals.

### Photocatalytic hetero-dihalogenation of alkenes using 1,2-dibromoethane and 1,1,2,2-tetrachloroethane (general procedure 4)

A 10 mL quartz tube was equipped with a stir bar and then filled with the corresponding alkenes (0.1 mmol, 1.0 equiv.), 1,2-dibromoethane (0.17 mmol, 1.7 equiv.), 1,1,2,2-tetrachloroethane (0.95 mmol, 9.5

equiv.), and 5 mg of Cu-CsPbBr$_3$ QDs. Then, 1 mL of acetonitrile was added. After the respective reaction time, the subsequent steps followed the same procedure as in **general procedure 3**.

## TA measurements

The femtosecond absorption spectra measurements were conducted using a laser amplifier (Coherent Astrella) and a pump-probe TA spectrometer (Helios, Ultrafast System). The 800 nm fundamental beam was divided into two beams, with one beam generating the pump pulse at 350 nm and the other beam focused through a 1 mm CaF$_2$ crystal to generate a white light probe pulse spanning from 400 to 600 nm. In a typical TA experiment, the QD sample dispersed in hexane was excited by the 350 nm pump beam, and the corresponding optical density of the sample was recorded across a wavelength range of 400–600 nm at various time delays. To study the effects of 1,2-dibromoethanes or 1,1,2,2-tetrachloroethanes, a specific amount of these compounds was added to the QD colloid sample, and the optical density of the samples was recorded under identical conditions.

## Data availability

The data that support the findings of this study are available from the corresponding author upon reasonable request. Source data are provided in this paper.

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

## Acknowledgements

We acknowledge the financial support from the National Key R&D Program (2021YFB4000600), the National Natural Science Foundation of China (22022406), the Natural Science Foundation of Tianjin (20JCJQJC00110 and 20JCYBJC00590), the 111 project (B12015), and the Haihe Laboratory of Sustainable Chemical Transformations. We would also like to express our appreciation to Yanfang Hu for their assistance with HRTEM characterization and to Zhang San from Shiyanjia Lab for conducting the XPS measurement.

## Author contributions

W.X. and Y.L. conceived and designed the experiments. Y.L. and Y.G. conducted the photocatalytic reactions. Z.D. provided valuable suggestions for organic synthesis. Y.C. performed the purification of the products. Y.L., Y.G., and T.W. carried out the synthesis and characterizations of the QD photocatalysts. Y.L. and Y.W. contributed to the data analysis. M.Y. provided important insights for QD characterizations. The paper was written by W.X. and Y.L. with contributions from all authors. C.Z. reviewed and revised the paper.

## Competing interests

The authors declare no competing interests.
