## [Peer Review File · Nature Communications]

REVIEWER COMMENTS

Reviewer #1 (Remarks to the Author):

The authors report in their manuscript the development of Perovskite CsPbBr₃ quantum dots and their use in visible light photocatalysis for vicinal dehalogenation reactions of alkenes and alkynes. First, the material is briefly introduced and analyzed by spectroscopic methods. The following part then describes the photocatalytic application of the material in first dibromination, then dichlorination and lastly hetero-dihalogenation reactions.

Although the reaction definitely represents an interesting approach, the mechanism seems to be different than reported by the authors. Both chlorine and bromine radicals are known hydrogen atom transfer mediators that have been used for allylic C-H abstraction (cyclohexene, 11a) and aldehydes (24a) among others. The postulated mechanism would therefore anticipate different products to be formed than those observed. Since the concentration of the halogen radicals are rather low, Wohl-Ziegler reactions would be expected. Could this be observed in any case, e.g. for cyclohexene? The observed products rather indicate that X₂ (X = Br, Cl) is the halogenation reagent.

Line 198-204: I assume the author wants to refer to a kinetic issue with the photocatalyst hot-state lifetime. This needs to be rephrased to more precisely point towards the issue and to avoid filler phrases like “too sluggish” or “smoothly extract”. The formed radical intermediate was reported to have a high concentration. Since electronically excited states are usually present in very low concentrations and the QY was determined to be 3.6%, this does not appear to be reasonable. Additional proofs have to be given that support the hypotheses described in the paragraph.

Minor comments:

- a) The ESI could benefit from a content table in the beginning.
- b) No statement about the reaction time was made. This should be included in the reaction scheme.
- c) Line 181: The term “high yields” usually expects yields larger than 70%, while even compounds 1a-7a only at best derived in 70% yield and should be rephrased.
- d) Line 186-190: It seems that heteroatoms with lower oxidation potentials (S, P) or reduction potentials (NO₂) derive in significantly lower yields and are not as well compatible as described by the authors with respect to other groups. The formulation should be adjusted.
- e) NMR spectra should be scaled completely to the top. The ppm-area should be unified and shown from at least -1 to 10 ppm for ¹H-NMR and -10 to 200 ppm for ¹³C-NMR.
- f) Product characterization appears to be only performed by NMR analysis. This is usually not sufficient and should be added by at least one other analytical method, e.g. mass spectrometry.

These findings extend previous methods significantly.

The mechanistic hypothesis needs further investigation.

Overall, in my opinion, the manuscript should be published after the addition of the suggested material.

Reviewer #3 (Remarks to the Author):

Xie and coworkers report a photochemical method for vicinal dihalogenation using a lead halide perovskite quantum dots as catalysts. A redox shuttle approach is used similar to previous work by Morandi and Waldvogel (reference 10 in the author's manuscript) which used an electrochemical approach instead of photochemical approach. In this work the halide source is 1,1,2,2-tetrachloroethane for the dichlorination. A Cu-containing lead-quantum dot was identified to enable the hetero-dihalogenation of alkenes in combination with 1,2-dibromoethane as the bromination source. Time-resolved spectroscopy and various radical trapping experiments provide insight into the reaction mechanism. While the transformation is not novel, the catalyst able to do this transformation provides novelty. Additionally the catalytic efficiency is high in terms of the turn over number (TON = 120,000 for the vicinal dichlorination).

Several drawbacks of this method are:

(1) The substrate scope for the three methods presented (Figures 3c, 4c, and 5c) are fairly narrow. For example, the substrate scope for the method presented in Figure 4c and 5c lack heterocycles or a broad range of functional groups. Nor do the structures showcase complexity (e.g. no stereocenters are present and most compounds have low molecular weight (<240 g/mol)). Figure 4c (scope for vicinal dichlorination) only has an ester as a functional group in the scope while the rest are hydrocarbon or halogenated hydrocarbons. In order to have major impact on synthetic organic chemistry functional groups like amines, alcohols, ethers, amides, heterocycles (pyridine, pyrimidine, pyrazole, indole, imidazole, furan, thiophene) should be demonstrated. The lack of these functional groups in the scope either suggests the method has narrow scope or is not yet fleshed out. A similar observation can be made with regards to the hetero dehalogenation scope illustrated in Figure 5c. Only one heterocycle (product 21c) is present in the scope shown in Figure 3c (debromination) and its yield is rather low (20%).

(2) The yields are low to moderate at best (the majority of the yields are in the 30 to 55% range).

(3) The use of lead in the catalyst is not environmentally friendly, both in terms of the waste generated as well as the toxicity and hazards associated with working with lead quantum dots.

(4) It is unclear what advantage this method provides compared to existing methods.

The manuscript may consider citing the following related references: (a) *Angew. Chem. Int. Ed.* 2022, 62, e202213630, (b) *J. Org. Chem.* 2022, 87, 9551, (c) *Chem. Rev.* 2022, 122, 2487.

REVIEWER COMMENTS

Reviewer #1 (Remarks to the Author):

The authors report in their manuscript the development of Perovskite CsPbBr₃ quantum dots and their use in visible light photocatalysis for vicinal dehalogenation reactions of alkenes and alkynes. First, the material is briefly introduced and analyzed by spectroscopic methods. The following part then describes the photocatalytic application of the material in first dibromination, then dichlorination and lastly hetero-dihalogenation reactions.

Although the reaction definitely represents an interesting approach, the mechanism seems to be different than reported by the authors. Both chlorine and bromine radicals are known hydrogen atom transfer mediators that have been used for allylic C-H abstraction (cyclohexene, 11a) and aldehydes (24a) among others. The postulated mechanism would therefore anticipate different products to be formed than those observed. Since the concentration of the halogen radicals are rather low, Wohl-Ziegler reactions would be expected. Could this be observed in any case, e.g. for cyclohexene? The observed products rather indicate that X₂ (X = Br, Cl) is the halogenation reagent.

Response: We thank the reviewer for her/his constructive suggestions which are very helpful for improving the quality of our work. Accordingly, we conducted additional experiment using white LED irradiation on a reaction mixture containing cyclohexene (0.1 mmol, 1.0 equiv.), 1,2-dibromoethane (DBE, 0.5 mmol, 5.0 equiv.) and CsPbBr₃ QDs (4 mg) in 1 mL of dichloromethane (DCM). This resulted in the formation of the primary dibromination products (11a, as depicted in Fig. R1) and a trace amount of brominated products (11a'). These findings indicate that the Wohl-Ziegler reactions are not the predominant pathways in this work. Furthermore, the introduction of 2,2,6,6-tetramethylpiperidine-1-oxyl (TEMPO), a radical trapping agent, was observed to hinder the formation of dibrominated products. High-resolution mass spectrometry data in Fig. R2 reveal that key radical species, such as bromine and carbon-centered radicals, were effectively captured by the trapping agent. The radical trapping experiment serves as a strong evidence to support the presence of radical pathways in the reaction mechanism.

Fig. R1. Photocatalytic shuttle dibromination reaction using cyclohexene reactant.

Fig. R2. Radical trapping experiment of the dibromination reaction.

Line 198-204: I assume the author wants to refer to a kinetic issue with the photocatalyst hot-state lifetime. This needs to be rephrased to more precisely point towards the issue and to avoid filler phrases like “too sluggish” or “smoothly extract”. The formed radical intermediate was reported to have a high concentration. Since electronically excited states are usually present in very low concentrations and the QY was determined to be 3.6%, this does not appear to be reasonable. Additional proofs have to be given that support the hypotheses described in the paragraph.

Response: The kinetic issue of hot electron transfer has been rephrased in the revised

manuscript:

“The photocatalytic dichlorination experiments are carried out using styrene as the archetypal substrate, CsPbBr₃ QDs as the photocatalysts and acetonitrile (MeCN) as the solvent instead of DCM in dibromination protocol in order to exclude other chlorination source. However, only 5% of 1,2-dichloride 1b is detected by GC, despite of a >95% consumption of styrene (see Route II in Fig. 4a). A 1,1,2-trichloroethane-centered radical generated by the first single-electron reduction of TCE can undergo an undesired transformation with styrene to yield by-products 1b' via an atom transfer radical addition (ATRA) reaction (Supplementary Fig. 9). In QD-catalyzed dichlorinations, the 1,1,2-trichloroethane-centered radicals fail to extract the second hot electron from CsPbBr₃ QDs to further release a second Cl⁻ anion, which leads that the second hot electron transfer process cannot outpace competing ATRA side reactions.”

The addition of TEMPO is found to inhibit the formation of dichlorination products. High-resolution mass spectrometry indicates that the 1,1,2-trichloroethane-centered radical species are trapped by the agent, but •Cl radicals are not detected (Fig. R3). Therefore, we speculate that carbon-centered radicals show relative higher concentration than •Cl radicals, despite that they are both present in very low concentrations. This is because the generation of the •Cl radicals involves a consecutive photoredox cascade process.

Fig. R3. Radical trapping experiment and generation process of the key radicals in the dichlorination reaction.

Minor comments:

a) The ESI could benefit from a content table in the beginning.

Response: We thank the reviewer for the helpful suggestion. A content table has been added in the revised SI.

b) No statement about the reaction time was made. This should be included in the reaction scheme.

Response: The reaction time has been schemed in Fig. 3 to Fig. 5 in the revised manuscript.

c) Line 181: The term “high yields” usually expects yields larger than 70%, while even compounds 1a-7a only at best derived in 70% yield and should be rephrased.

Response: According to the reviewer’s comment, “high yields” has been changed to “moderate to good yields”.

d) Line 186-190: It seems that heteroatoms with lower oxidation potentials (S, P) or reduction potentials (NO₂) derive in significantly lower yields and are not as well compatible as described by the authors with respect to other groups. The formulation should be adjusted.

Response: By employing CsPbBr₃ QDs as photocatalysts, thioether-functionalized alkenes undergo conversion to the corresponding vicinal dibromides in 30~40% low yields (30a in Fig. R4). The lower yields could be explained by some undesired transformations of the alkenes. A new alkene containing sulfur atom was designed as substrates to obtain the corresponding 1,2-dibromide in ~60% yield (31a in Fig. R4). To further increase the yields of other substrates with lower reduction or oxidation potentials, a dibromination protocol is developed using Cu-CsPbBr₃ QDs (0.00049mol%) as the photocatalysts and MeCN as solvent, under nitrogen atmosphere at 25 °C and illuminated by a 10 W white LED. To our delight, the low yield (37%) of the substrate with aldehyde group (23a in Fig. 4), which is sensitive to oxidation, is increased to moderate yield (60%). Two alkenes with nitro group (22a and 24a in Fig. R4) were used as substrates to produce the corresponding 1,2-dibromide in 65% and 84% yields, respectively. However, alkene with phosphorus atom is not as well compatible in the dibromination protocol because of the oxidation of phosphorus. Therefore, this alkene substrate has been removed from Fig. 3.

Fig. R4. Scope of alkene with functional group sensitive to oxidation and reduction of the dibromination reactions.

e) NMR spectra should be scaled completely to the top. The ppm-area should be unified and shown from at least -1 to 10 ppm for ¹H-NMR and -10 to 200 ppm for ¹³C-NMR.

Response: Thank the reviewer for the suggestion. Accordingly, the NMR spectra have been revised.

f) Product characterization appears to be only performed by NMR analysis. This is usually not sufficient and should be added by at least one other analytical method, e.g. mass spectrometry.

Response: According to the reviewer's comments, MS has been added for product characterization. Please see the revised supplementary information.

These findings extend previous methods significantly.

The mechanistic hypothesis needs further investigation.

Overall, in my opinion, the manuscript should be published after the addition of the suggested material.

Response: We thank the reviewer for the positive comments and constructive suggestions on our study. We have revised our manuscript and supplementary information very carefully and included additional experimental data to support the mechanistic hypothesis. With the new version of the manuscript (changes are highlighted in yellow color), we hope the reviewer agree with us that the present work can be accepted for publication in Nature Communications.

Reviewer #3 (Remarks to the Author):

Xie and coworkers report a photochemical method for vicinal dihalogenation using a lead halide perovskite quantum dots as catalysts. A redox shuttle approach is used similar to previous work by Morandi and Waldvogel (reference 10 in the author's manuscript) which used an electrochemical approach instead of photochemical approach. In this work the halide source is 1,1,2,2-tetrachloroethane for the dichlorination. A Cu-containing lead-quantum dot was identified to enable the hetero-dihalogenation of alkenes in combination with 1,2-dibromoethane as the bromination source. Time-resolved spectroscopy and various radical trapping experiments provide insight into the reaction mechanism. While the transformation is not novel, the catalyst able to do this transformation provides novelty. Additionally the catalytic efficiency is high in terms of the turn over number (TON = 120,000 for the vicinal dichlorination).

Response: We thank the reviewer for the positive comments on our work using perovskite quantum dot (QD) catalysts for the high efficient redox shuttle reactions.

Several drawbacks of this method are:

(1) The substrate scope for the three methods presented (Figures 3c, 4c, and 5c) are fairly narrow. For example, the substrate scope for the method presented in Figure 4c and 5c lack heterocycles or a broad range of functional groups. Nor do the structures showcase complexity (e.g. no stereocenters are present and most compounds have low molecular weight (<240 g/mol)). Figure 4c (scope for vicinal dichlorination) only has an ester as a functional group in the scope while the rest are hydrocarbon or halogenated hydrocarbons. In order to have major impact on synthetic organic chemistry functional groups like amines, alcohols, ethers, amides, heterocycles (pyridine, pyrimidine, pyrazole, indole, imidazole, furan, thiophene) should be demonstrated. The lack of these functional groups in the scope either suggests the method has narrow scope or is not yet fleshed out. A similar observation can be made with regards to the hetero dehalogenation scope illustrated in Figure 5c. Only one heterocycle (product 21c) is present in the scope shown in Figure 3c (debromination) and its yield is rather low (20%).

Response: We appreciate the reviewer's insightful comments. To broaden the substrate scope, we have included an array of substrates with potential reactivity. With the assistance of students specializing in organic chemistry, we dedicated significant effort to synthesizing, separating and purifying these substrates (1e to 15e, as shown in Fig. R1). All substrates exhibited purities greater than 95% and were characterized by ¹H NMR (400 MHz, Chloroform-*d*; see characterization section). Subsequently, we have tested alkenes featuring various functional groups, including hydroxyl, ethers, amides, and heterocycles (pyridine, furan, pyrimidine, pyrazole, thiophene), as substrates for dihalogenation reactions. Employing

CsPbBr₃ or Cu-CsPbBr₃ QDs as photocatalysts, the alkenes containing pyridine, pyrimidine, thiophene, furan, pyrazole, and hydroxyl groups were successfully transformed into the corresponding vicinal dibromides (28a to 33a in Fig. R2) with moderate to good yields.

Fig. R1. Synthesized alkenes with various functional groups.

Fig. R2. Scope of alkene of the photocatalytic shuttle dihalogenation reactions. Dibromination reaction conditions: 0.1 mmol alkene compound, 0.5 mmol DBE, 4 mg Cu-CsPbBr₃ QD catalyst; dichlorination reaction conditions: 0.1 mmol alkene compound, 0.5 mmol TCE, 4 mg

Cu-CsPbBr₃ QD catalyst; hetero-dihalogenation reaction conditions: 0.1 mmol alkene compound, 0.17 mmol DBE, 0.95 mmol TCE, 5 mg Cu-CsPbBr₃ QD catalyst, 1 mL of MeCN and irradiation using a 10 W white LED at 25 °C. Isolated yields for dibrominated products. ¹H-NMR yields for dichlorinated and hetero-dihalogenated products. *CsPbBr₃ QD as catalyst and DCM as solvent.

Following the dichlorination and hetero-dihalogenation protocols respectively, substrates containing ethers, thiophene, pyridine, amides, and pyrimidine were converted to 1,2-dichlorides (14b to 16b) and 1,2-dihalides (17c to 19c, as shown in Fig. R2). The low yields can be attributed to undesired ATRA, oxidation, and unidentified reactions. During the chlorination process, a small quantity of Cl⁻ anions is initially generated in the reduction half-reaction of TCE on the perovskite surface. Subsequently, the Cl⁻ anions are transferred to the perovskite lattice, forming a new CsPbCl₃ crystalline phase via ion exchanges. Since CsPbCl₃ exhibits a lower extinction coefficient compared to CsPbBr₃ in the visible range, the remaining Cl⁻ (E_{ox} = 1.05 V vs. SCE) are slowly oxidized at the oxidizing sites, generating •Cl radicals at low concentrations. At present, there is no effective strategy to prevent such anion exchanges. Fig. R3 shows some other unsuccessful examples. Such substrates exhibited incompatibility with the reaction system (1d to 20d), providing evidence of strong photosensitivity, insufficient photoredox reactivity, undesired side reactions, or other reasons.

Fig. R3. Unsuccessful examples of alkene of the photocatalytic shuttle dihalogenation reactions.

(2) The yields are low to moderate at best (the majority of the yields are in the 30 to 55% range).

Response: In order to boost the dibromination yields, a series of transition metals are incorporated into CsPbBr₃ QDs by a cation-exchange method (Fig. R4). After several experiments, a dibromination protocol is developed using Cu-CsPbBr₃ QDs (0.00049 mol%) as the photocatalysts, DBE as the halogen source and MeCN as solvent, under nitrogen atmosphere at 25 °C and illuminated by a 10 W white LED. After 24 h illumination, the reaction yields 80% (1,2-dibromoethyl)benzene (**1a**). Using this new dibromination protocol, low and moderate yields are increased to moderate and even good yields (see Fig. 3).

However, two hot electron transfer is more challenging in visible-light-driven dichlorination reactions because a 1,1,2-trichloroethane-centered radical generated by the first single-electron reduction of 1,1,2,2-tetrachloroethane (TCE) is too stable to extract the second hot electron from CsPbBr₃ QDs. A series of metal catalysts are incorporated into QDs to accelerate the hot electron transfer, unfortunately, the dichlorination yields are moderate at best.

Fig. R4. Survey of metal catalysts for the dibromination and dichlorination reactions.

Conventional optimization method of reaction condition is tuning the external environment of the catalysts. Owing to their intrinsic chemical instability issues, perovskite QDs undergo destruction under external stimulation such as strong polar solvent, acid, base, coordinating anions, and temperature. Therefore, we intend to further change the interfacial environment of the perovskite QD catalysts for increase the dichlorinated yields. According to the literatures (*Nat. Nanotechnol.* **2023**, 18, 160–167; *ACS Catal.* **2021**, 11, 14284–14292; *Nano Energy* **2021**, 80, 105532), carbon materials have been recognized as prospective catalysts for the electrocatalytic 1,2-dichloroethane (DCE) dechlorination reaction. Inspired by the literatures, we use various carbon-based nanomaterials including g-C₃N₄, carbon nanotubes, carbon nanosheets, and carbon black as Cu-QD support materials to accelerate

electron transfer in photocatalytic process. Via an ultrasonication-assisted self-assembly process, Cu-CsPbBr₃ QDs are intended to be decorated on the surface of these carbon nanomaterials. Using the heterostructured photocatalysts, however, the yields of the dichlorinated reactions are unable to be further increased (see Fig. R5). Although these carbon nanomaterials show excellent electrical conductivity, they fail to promote the transmission of hot charge carriers in Cu-CsPbBr₃ QDs. Therefore, much exploring works are needed to develop perovskite-based heterostructures for the transfer dichlorination reaction with high yields, which is our direction for the future work.

Fig. R5. Survey of support materials for the dichlorination reaction.

(3) The use of lead in the catalyst is not environmentally friendly, both in terms of the waste generated as well as the toxicity and hazards associated with working with lead quantum dots.

Response: The reviewer is right. Lead is toxic and will result in serious environmental concerns or human health related issues when handling Pb-based perovskites without sufficient protection. Inspired by the reviewer's comment, we attempted to use lead-free quantum dot catalysts in our reactions and synthesized colloidal Cs₂AgBiBr₆ nanocrystals according to literature (*Chem. Mater.* **2019**, 31, 7962–7969). The corresponding optical and morphological characterizations for Cs₂AgBiBr₆ QDs are shown in Fig. R6.

A reaction vial containing styrene as substrates, Cs₂AgBiBr₆ QDs as the photocatalysts, DBE as the halogen source, and DCM as solvent, under nitrogen atmosphere at 25 °C, was illuminated by a 10 W white LED. However, after 36 h, trace yield of dibrominated products was obtained. The low optical activity of the lead-free perovskite QDs in the visible light region is a potential reason for their low photocatalytic performance in this dibromination reaction. Limited by time, the lead-free QD catalysts are not available in our shuttle dehalogenation reactions at present. We will continue to look for ways to replace the lead quantum dots with environmental friendly catalysts.

Fig. R6. Visible-light-driven dibromination reactions using Cs₂AgBiBr₆ QDs as photocatalysts and the characterizations of Cs₂AgBiBr₆ QDs.

(4) It is unclear what advantage this method provides compared to existing methods.

Response: In a pioneer work, an electrochemically-assisted shuttle (e-shuttle) paradigm to synthesize valuable dihalogenated molecules was demonstrated by Morandi and Waldvogel group (*Science*, 2021, 371, 507–514). Compared with electrocatalysis, visible-light photoredox catalysis is more attractive for accessing the shuttle (p-shuttle) dihalogenation due to the solar energy input. Hence, we have developed a visible-light-driven reversible transfer dihalogenation reaction using perovskite quantum dot catalysts under room temperature, and this is the first example of efficiently utilizing the energetic hot electrons for inducing novel organic transformations.

The manuscript may consider citing the following related references: (a) *Angew. Chem. Int. Ed.* 2022, 62, e202213630, (b) *J. Org. Chem.* 2022, 87, 9551, (c) *Chem. Rev.* 2022, 122, 2487.

Response: According to the reviewer's suggestion, the related references have been reasonably cited in the revised manuscript.

4-vinylphenyl 3-(pyridin-3-yl)propanoate (**1e**)

4-vinylphenyl isonicotinate (**2e**)

4-vinylphenyl 6-methoxynicotinate (**3e**)

N-methyl-4-vinylbenzamide (**4e**)

N-(4-vinylphenyl)acetamide (**5e**)

4-vinylphenyl 1,5-dimethyl-1H-pyrazole-4-carboxylate (**6e**)

4-vinylphenyl 1H-indole-2-carboxylate (**7e**)

4-vinylphenyl furan-2-carboxylate (**8e**)

4-vinylphenyl thiophene-2-carboxylate (**9e**)

N-(3-vinylphenyl)acetamide (**10e**)

4-vinylphenyl pyrimidine-5-carboxylate (**11e**)

3-vinylphenyl 2-(pyrimidin-2-yl)acetate (**12e**)

4-vinylphenyl imidazo[1,2-a]pyridine-2-carboxylate (**13e**)

1-(4-vinylphenyl)-1H-imidazole (**14e**)

REVIEWER COMMENTS

Reviewer #1 (Remarks to the Author):

The group of Wei Xie reported a method for the photocatalytic vicinal dihalogenation of alkenes using lead halide Perovskite Quantum Dot catalysts inspired by the previous work of Morandi and Waldvogel in 2021. They state that the mechanism occurs via transfer of hot electrons generated in the metal quantum dot's band structure to the substrate leading to the desired reduction from halide anions to the respective radicals. Depending on the stoichiometry between styrene derivative and bromo-/chloro alkene, dehalogenation of styrene together with ethene formation or reduction of the halogenated alkane starting materials is achieved. The driving force for the reactions is supposed to be the formation of ethene.

In the manuscript the expressions "hot" or "cold" electrons are used rather often in combination with the reactivity of the quantum dots. In order to achieve a better systematic and scientific understanding for the interesting chemistry of QD the authors should explain those terms in more detail. Additionally, they state that the LUMO of the dibromo alkane species is too high (or in general higher than the conduction band minimum of the QD). What does this statement exactly mean? Does this energy have an impact on the energetic profile of this reaction or are those dihalogenated alkanes in general difficult to react?

Besides many different characteristics of the quantum dots an increasing kinetic energy of electrons is mentioned. How is this energy related to the overall electron transfer or photocatalytic transformation? Is it advantageous or disadvantageous? It is inevitable to explain specific research fields in a more basic and simple way to guarantee understanding of a heterogenous group of readers.

Figure 2 in the supporting information shows transient absorption spectra performed at 480 and 525 nm. Why were those wavelengths chosen for spectroscopical investigations but the photoreactions themselves were run under irradiation with white light? In case the spectroscopical data were accurate at those wavelengths the authors should think about defining a specific wavelength for their dihalogenation reaction as well. In general, in both, the manuscript as well as in the supporting information several optimization tables are missing such as light source screening, reaction kinetics, reaction time optimization, catalyst loading screening and solvent volume survey (this might have an impact on the performance of the photocatalyst as it gets closer to the light source).

The group of Xie should also explain experimental procedures more accurately. Do the QD perfectly dissolve in the mentioned solvent systems or is scattering a problem? In the SI the researchers perform spectroscopy in colloidal systems, how does this work exactly? Does the amount of halide anions on the surface of the quantum dots change the solubility of the catalyst (in DCM), can precipitation be a problem at a given time?

Ethene is the side product of the reaction that shows good atom economy. As some of the ethene molecules might remain in liquid phase instead of fully diffusing into the gas phase is a 2+2 cycloaddition between styrene and the ethene a possible side reaction?

In the supporting information different analytical methods are listed. Among them ex-situ PL is used to evaluate the time-dependant halide conversion on the quantum dots in presence/absence of the styrene derivative. The authors should explain the principle of this spectroscopical method in more detail.

The proton NMR spectra show a lot of impurities from chromatography solvents in the region of 0-2 ppm. To have the adequate quality for Nature Communication the compounds should definitely be purified again, or the residual solvent impurities have to be removed under reduced pressure. The researchers should mention the exact solvent stoichiometry of petrol ether and ethyl acetate for chromatography to achieve good reproducibility for readers.

In general, interesting chemistry is presented in the manuscript of the Xie group. The transformation shows high atom economy and the overall methodology using tuneable quantum dots with interesting (photo-)physical properties allows for many different applications although the principle is not new. The scientific level of the manuscript is impressively high but at the same time the background theory is difficult to understand due to complex sentence structure and the lack of proper scientific explanations of expression in the field of QD photocatalysis. Additionally, data evaluation, such as presenting NMR spectra (and their purity) should be improved, and missing experiments need to be added. After revision of the manuscript the work is suitable for Nature Communications.

Reviewer #2 (Remarks to the Author):

The revised manuscript is suitable for publication. No additional comments beyond statements already made in first round of reviewing.

Point-by-point response to the reviewers' comments

Reviewer #1 (Remarks to the Author):

The group of Wei Xie reported a method for the photocatalytic vicinal dihalogenation of alkenes using lead halide Perovskite Quantum Dot catalysts inspired by the previous work of Morandi and Waldvogel in 2021. They state that the mechanism occurs via transfer of hot electrons generated in the metal quantum dot's band structure to the substrate leading to the desired reduction from halide anions to the respective radicals. Depending on the stoichiometry between styrene derivative and bromo-/chloro alkene, dehalogenation of styrene together with ethene formation or reduction of the halogenated alkane starting materials is achieved. The driving force for the reactions is supposed to be the formation of ethene.

In the manuscript the expressions “hot” or “cold” electrons are used rather often in combination with the reactivity of the quantum dots. In order to achieve a better systematic and scientific understanding for the interesting chemistry of QD the authors should explain those terms in more detail. Additionally, they state that the LUMO of the dibromo alkane species is too high (or in general higher than the conduction band minimum of the QD). What does this statement exactly mean? Does this energy have an impact on the energetic profile of this reaction or are those dihalogenated alkanes in general difficult to react?

Response: We thank the reviewer for her/his constructive comments and suggestions, which are very helpful for improving the quality of our work. In general, hot electrons, which are electrons above the conduction band, exhibit higher reduction potential compared to cold electrons, which reside on the band edge (see Figure R1). Hot electrons, with $E < -1.3$ V vs SCE, carry large kinetic energies and possess a more favorable reduction potential. The reductive transformation of 1,2-dibromoethane (DBE) on the surface of quantum dots (QDs) involves two-step reduction processes that generate two X^- anions and an alkene. However, the DBE molecule, with reduction potentials $E_1 = -0.87$ V and $E_2 = -2.32$ V vs SCE, possesses a higher reduction potential than the conduction band minimum of QDs ($V_{CB} = -1.3$ V vs SCE).

Fig. R1. Schematics of hot electron transfer in the shuttle reactions.

Besides many different characteristics of the quantum dots an increasing kinetic energy of electrons is mentioned. How is this energy related to the overall electron transfer or photocatalytic transformation? Is it advantageous or disadvantageous? It is inevitable to explain specific research fields in a more basic and simple way to guarantee understanding of a heterogenous group of readers.

Response: According to the schematics of hot electron transfer in the shuttle reactions shown in Figure R1, only hot electrons possessing substantial kinetic energies can overcome the potential barrier and be efficiently transferred to DBE, which is facilitated by their favorable energy match. We believe that these high-energy electrons are advantageous for facilitating the challenging reactions. We have included corresponding explanations in more simple way in the revised manuscript.

Figure 2 in the supporting information shows transient absorption spectra performed at 480 and 525 nm. Why were those wavelengths chosen for spectroscopical investigations but the photoreactions themselves were run under irradiation with white light? In case the spectroscopical data were accurate at those wavelengths the authors should think about defining a specific wavelength for their dihalogenation reaction as well. In general, in both, the manuscript as well as in the supporting information several optimization tables are missing such as light source screening, reaction kinetics, reaction time optimization, catalyst loading screening and solvent volume survey (this might have an impact on the performance of the photocatalyst as it gets closer to the light source).

Response: We thank the reviewer for this important suggestion. Upon light illumination, the CsPbBr₃ QDs exhibit distinct photoinduced absorption bands at 480 nm and 525 nm in their TA spectra, which are inherent to the catalyst material and not specifically determined by the wavelengths used in our experiment. For the photocatalysis experiment, we utilized white light spanning the wavelength range from 400 to 700 nm, thereby covering both of these absorption bands.

According to the reviewer's suggestion, we carried out several additional optimization experiments to investigate various aspects including light source selection, reaction kinetics, reaction time optimization, catalyst loading screening, and solvent volume using dibromination reactions as model systems. The results of these comprehensive photocatalytic experiments are presented in Tables R1-4 and Fig. R2.

Table R1. Survey of excitation wavelengths.

Entry	Wavelength [nm]	GC Yield [%]
1	380	23
2	420	24
3	455	55
4	500	30
5	590	0

Table R2. Survey of reaction times.

Entry	Reaction time [h]	GC Yield [%]
1	6	27
2	12	42
3	24	63
4	36	60
5	48	62

Table R3. Survey of catalyst loadings.

Entry	Catalyst loading [mg]	GC Yield [%]
1	0	0
2	1	65
3	2	59
4	4	52

Table R4. Survey of solvent volumes.

Entry	Solvent volume [mL]	GC Yield [%]
1	0.5	37
2	1	58
3	2	57
4	4	64
5	8	60

Fig. R2. Kinetic data obtained for the dibromination reaction using CsPbBr₃ QDs as catalysts.

The group of Xie should also explain experimental procedures more accurately. Do the QD perfectly dissolve in the mentioned solvent systems or is scattering a problem? In the SI the researchers perform spectroscopy in colloidal systems, how does this work exactly? Does the amount of halide anions on the surface of the quantum dots change the solubility of the catalyst (in DCM), can precipitation be a problem at a given time?

Response: The stability of the QD catalysts is an important issue. According to the Tyndall effect (Fig. R3) of the reaction system, the CsPbBr₃ QD catalysts are confirmed to be in the form of colloid dispersed in DCM. After approximately 30 minutes of illumination under white LED, the QDs exhibit remarkable colloidal stability in the solvent (see Fig. R3), although they have been transformed into CsPbCl₃ via surface anion-exchange. Therefore, the QDs maintain their colloidal form throughout the spectroscopy characterization process, and no precipitation was observed in DCM.

Fig. R3. Photograph of CsPbBr₃ QDs before and after LED irradiation.

Ethene is the side product of the reaction that shows good atom economy. As some of the ethene molecules might remain in liquid phase instead of fully diffusing into the gas phase is a 2+2 cycloaddition between styrene and the ethene a possible side reaction?

Response: Thank the reviewer for this comment. Accordingly, we conducted additional experiment using white LED irradiation on a reaction mixture containing styrene (0.1 mmol, 1.0 equiv.), 1,2-dibromoethane (DBE, 0.5 mmol, 5.0 equiv.) and CsPbBr₃ QDs (4 mg) in 1 mL of dichloromethane (DCM). As shown in Fig. R4, the cycloaddition product, phenylcyclobutane, was not detected in the reaction mixture. We think that the current reaction environment inhibits the cycloaddition side reaction.

Fig. R4. GC graphs of dibromination mixture and pure phenylcyclobutane.

In the supporting information different analytical methods are listed. Among them ex-situ PL

is used to evaluate the time-dependant halide conversion on the quantum dots in presence/absence of the styrene derivative. The authors should explain the principle of this spectroscopical method in more detail.

Response: The emission wavelength of excitonic PL in CsPbX₃ perovskite QDs is strongly influenced by their chemical composition. The CsPbCl₃ QDs exhibit an intrinsic exciton PL peak at approximately 410 nm (blue light), while the CsPbBr₃ QDs display an intrinsic exciton PL peak at around 520 nm (green light). On the other hand, the CsPbBr_xCl_{3-x} QDs exhibit intrinsic exciton PL across a wavelength range spanning from 410 to 520 nm. When halide exchange occurs on the surface of QDs, a continuous shift in the PL wavelength can be observed. The principle of this ex-situ PL method has been discussed and included in supplementary information.

The proton NMR spectra show a lot of impurities from chromatography solvents in the region of 0-2 ppm. To have the adequate quality for Nature Communication the compounds should definitely be purified again, or the residual solvent impurities have to be removed under reduced pressure. The researchers should mention the exact solvent stoichiometry of petrol ether and ethyl acetate for chromatography to achieve good reproducibility for readers.

Response: We appreciate the reviewer's helpful suggestion. In order to improve the quality of the ¹H-NMR spectra, we have invested considerable efforts in synthesis, separation and purification of the target products through additional experiments. Fig. R5 shows the ¹H NMR spectra of a typical compound after removing the solvent impurities. Additionally, the stoichiometry of petrol ether and ethyl acetate has been included in supplementary information. Please refer to the revised supplementary information for further details.

Fig. R5. ^1H NMR (400 MHz, chloroform- d) of 1-chloro-4-(1,2-dibromoethyl)benzene before and after removing the solvent impurities.

In general, interesting chemistry is presented in the manuscript of the Xie group. The transformation shows high atom economy and the overall methodology using tuneable quantum dots with interesting (photo-)physical properties allows for many different applications although the principle is not new. The scientific level of the manuscript is impressively high but at the same time the background theory is difficult to understand due to complex sentence structure and the lack of proper scientific explanations of expression in the field of QD photocatalysis. Additionally, data evaluation, such as presenting NMR spectra (and their purity) should be improved, and missing experiments need to be added. After revision of the manuscript the work is suitable for Nature Communications.

Response: We thank the reviewer for the positive comments and constructive suggestions to improve the quality of our work. Accordingly, we have performed additional experiments and revised our manuscript and supplementary information very carefully. The new results are mainly included in the revised supplementary information. With the new version of the manuscript & supplementary information (changes are highlighted in yellow color), we hope the reviewer agree with us that the present work can be accepted for publication in *Nature Communications*.

Reviewer #2 (Remarks to the Author):

The revised manuscript is suitable for publication. No additional comments beyond statements already made in first round of reviewing.

Response: We thank the reviewer for the recommendation that our revised manuscript is suitable for publication.